# Differentially Private Iterative Screening Rules for Linear Regression

## Abstract

Linear $L_1$-regularized models have remained one of the simplest and most effective tools in data science. Over the past decade, screening rules have risen in popularity as a way to reduce the runtime for producing the sparse regression weights of $L_1$ models. However, despite the increasing need of privacy-preserving models for data analysis, to the best of our knowledge, no differentially private screening rule exists. In this paper, we develop the first private screening rule for linear regression. We initially find that this screening rule is too strong: it screens too many coefficients as a result of the private screening step. However, a weakened implementation of private screening reduces overscreening and improves performance.

## 1 Introduction

Sparse linear regression is an important statistical technique which can prevent overfitting and allow data analysts to better interpret model coefficients. Its applications are widespread, but it is especially useful when choosing features for further study or analyzing high-dimensional datasets with many features, such as those in medicine or finance (Kandel et al., 2013; Wang, 2010). Sparse regression is typically achieved through LASSO ($L_1$) optimization. In its constrained form, the LASSO optimization problem can be represented by

$$\widehat{\mathbf{w}} = \operatorname*{arg\,min}_{\mathbf{w} \in \mathbb{R}^d: \, \|\mathbf{w}\|_1 \leq \lambda} \frac{1}{n} \sum_{i=1}^n (y_i - \mathbf{w} \cdot \mathbf{x}_i)^2,$$

where $\lambda$ is the constraint parameter, $\mathbf{x}_1, \dots, \mathbf{x}_n \in \mathbb{R}^d$, and $y_1, \dots, y_n \in \mathbb{R}$. The Frank-Wolfe algorithm can be used to directly optimize over this objective (Frank & Wolfe, 1956).

However, the output $\widehat{\mathbf{w}}$ of this optimization may not be private. In other words, $\widehat{\mathbf{w}}$ can reveal information about specific training datapoints. This may be dangerous in fields like medicine and finance, where datapoints consist of sensitive information which should not be revealed when a model is used publicly (Khanna et al., 2022; Basu et al., 2021). To prevent private information leakage in cases like these, differential privacy can be used.

Differential privacy is a statistical technique which provides a guarantee of training data privacy. Specifically, given privacy parameters $\epsilon$ and $\delta$ and any two datasets $\mathcal{D}$ and $\mathcal{D}'$ differing on one datapoint, an approximate differentially private algorithm $\mathcal{A}$ satisfies $\mathbb{P}\left[\mathcal{A}(\mathcal{D}) \in O\right] \leq \exp\{\epsilon\}\mathbb{P}\left[\mathcal{A}(\mathcal{D}') \in O\right] + \delta$ for any $O \subseteq \operatorname{image}(\mathcal{A})$ (Dwork et al., 2014).[1] In simple terms, a differentially private algorithm's outputs do not reveal whether or not a specific datapoint was used in its training.

Making an algorithm differentially private requires adding noise to its intermediate steps or outputs. Regardless of where the noise is added, its scale must be proportional to the algorithm's sensitivity, or how much its outputs can change when one of its input datapoints is added or removed. If this change is measured with

---

[1] An important consideration when employing differential privacy is the definition of two datasets *differing on one datapoint*. There are two ways to define this: one in which a datapoint from $\mathcal{D}$ is replaced with another to produce $\mathcal{D}'$, and another in which a datapoint is added or removed from $\mathcal{D}$ to produce $\mathcal{D}'$. In this paper, we use the latter definition for dataset adjacency.

an $L_1$ or $L_2$ norm, noise can be added with the Laplacian or Gaussian distributions, respectively. Further introductory details on differential privacy can be found in Near & Abuah (2021).

Differentially private high-dimensional regression optimization algorithms exist, but have flaws (Talwar et al., 2015; Wang & Gu, 2019). These optimization algorithms modify nonprivate sparse optimization techniques but can run slowly and produce dense or ineffective solutions. Additionally, many works on sparse differentially private regression discuss model selection algorithms which run prior to training (Kifer et al., 2012; Thakurta & Smith, 2013). However, these algorithms are computationally inefficient and can assume an exact level of sparsity of the final solution before choosing a support set prior to training.

In the nonprivate setting, sparsity on regression weights can be achieved by using screening rules. Screening rules are methods which discard features that do not contribute to a statistical model during training. They are most often used in $L_1$-regularized or $L_1$-constrained linear regression to set the coefficients of unimportant features to zero during the optimization process. In doing so, they improve the generalization of a model by counteracting overfitting and enable faster convergence. Screening rules have been used to improve the performance of sparse regression optimization on numerous datasets over the past decade and are even included in the popular `R` package `glmnet` (Wang et al., 2013; 2014; Raj et al., 2016; Ghaoui et al., 2010; Olbrich, 2015; Tibshirani et al., 2012; Friedman et al., 2021).

Unlike the aforementioned approaches for private regression, screening rules do not require a predetermined support set or level of sparsity. They efficiently check a mathematical condition to determine if a feature should be screened to 0, and are implemented with sparse optimizers during training to improve the rate of learning and stability.

A differentially private screening rule has the potential to combat overfitting and help private optimizers focus on a model's most important features. However, to the best of our knowledge, no differentially private screening rule exists. In this work, we privatize a screening rule used for convex problems developed by Raj et al. (2016), with the goal of testing whether such a modification can yield the benefits of screening rules while retaining accuracy and guarnateeing privacy. ~~In this work, we develop and explore a differentially private screening rule, and show two variants of its implementation.~~ We find that an aggresive implementation of the screening rule cannot accurately screen features, and we analyze why this is the case. We then create a weaker implementation strategy with improved results.

This paper is organized as follows. In Section 2, we review prior literature on sparse private regression and a relevant nonprivate screening rule. In Section 3, we find the sensitivity of this screening rule, and Section 4 shows how the rule can be tentatively implemented to produce sparsity. However, the screening rule presented in Section 4 is too strong, so Section 5 presents a way to weaken the screening rule, and Section 6 demonstrates experiments with this screening rule, showing that it can produce better results.

## 2    Related Work

Methods to produce private sparse regression weights all suffer in performance due to the addition of noise. Private $L_1$ optimizers must add higher levels of noise when run for more iterations, incentivizing practitioners to run fewer iterations (Talwar et al., 2015; Wang & Zhang, 2020). However, running an optimizer for fewer iterations means that the model will be limited in its learning. On the other hand, private model selection algorithms are computationally inefficient and run prior to training, meaning they are unable to reap the benefit of any information contained within partially trained coefficients of the weight vector (Lei et al., 2018; Thakurta & Smith, 2013). Although noise is necessary for privacy, an effective private screening rule would run with a private optimizer and improve the optimizer's performance by setting the coefficients of irrelevant features to 0. By using the screening rule on the current weight vector, it can adapt to the optimizer's updates and screen features more accurately.

To develop a differentially private screening rule, we adapt Theorem 11 of Raj et al. (2016)'s rule which is flexible to solving many types of regression problems. While other screening rules exist, they are geometry- and problem-specific (Ghaoui et al., 2010; Wang et al., 2014; 2013). **Our goal is to utilize Raj et al.**

**(2016)'s screening rule for $L_1$-constrained regression to induce sparsity on Talwar et al. (2015)'s $L_1$-constrained private Frank-Wolfe algorithm.**[2]

Since we use the private Frank-Wolfe algorithm (`DP-FW`), we also review it here. `DP-FW` uses the Frank-Wolfe method for $L_1$-constrained convex optimization, which chooses a vertex of the feasible region (scaled $L_1$ ball) which minimizes a linear approximation of the loss function. By doing this for $T$ iterations with appropriate step sizes, the algorithm satisfies $\mathcal{L}(\mathbf{w}^{(T)}) - \min_{\mathbf{w}^* \in \mathcal{C}} \mathcal{L}(\mathbf{w}^*) \leq \mathcal{O}(\frac{1}{T})$ (Frank & Wolfe, 1956; Jaggi, 2013). To privatize the Frank-Wolfe algorithm, Talwar et al. (2015) restrict the $L_\infty$ norm of datapoints so they can calculate the exact sensitivity of the gradients. They then use the report-noisy-max mechanism to noisily choose which component of the weight vector to update. Unfortunately, due to inexact optimization caused by the noisy selection process, the private Frank-Wolfe algorithm produces dense results, limiting its ability to be used in high-throughput or interpretability-restricted applications of regression. Despite this limitation, the Frank-Wolfe algorithm is ideal for an application of Raj et al. (2016)'s screening rule. Of current methods for private high-dimensional regression, recently summarized by Khanna et al. (2024), the Frank-Wolfe algorithm is unique in that it uses $L_1$-constrained optimization with updates to one component of the weight vector per iteration. Each of these conditions is important. The first is necessary because Raj et al. (2016)'s screening rule requires optimization over an $L_1$-constrained set. The second is important because if a private (noisy) optimization algorithm updates all components of a weight vector at each iteration, then any sparsity induced by applying a screening rule would be lost at the next iteration of optimization, since the output of the optimization would be dense.

To the best of our knowledge, this is the first work considering a differentially private screening rule. The difficulty of DP screening is counteracted by the reward of obtaining sparse *and* private regression, as normal DP destroys sparsity via the addition of noise.

## 3 Privatizing Iterative Screening

Raj et al. (2016) consider the problem $\min_{\mathbf{w} \in \mathcal{C}} f(\mathbf{X}\mathbf{w})$, where $\mathcal{C}$ is the feasible set of solutions and $f$ is $L$-smooth and $\mu$-strongly convex. They also define $\mathbf{x}_{(i)} \in \mathbb{R}^n \mathbf{x}_{(i)}$ to be the $i^{\text{th}}$ column of the design matrix

$$\mathbf{X} = \begin{bmatrix} \mathbf{x}_1 \\ \vdots \\ \mathbf{x}_n \end{bmatrix} \in \mathbb{R}^{n \times d} \mathbf{X}$$ and $\mathcal{G}_{\mathcal{C}}(\mathbf{w})$ to be the Wolfe gap function, namely $\max_{\mathbf{z} \in \mathcal{C}} (\mathbf{X}\mathbf{w} - \mathbf{X}\mathbf{z})^\top \nabla f(\mathbf{X}\mathbf{w})$. Given

this information, they prove that if

$$s_i = |\mathbf{x}_{(i)}^\top \nabla f(\mathbf{X}\mathbf{w})| + (\mathbf{X}\mathbf{w})^\top \nabla f(\mathbf{X}\mathbf{w}) + L(\|\mathbf{x}_{(i)}\|_2 + \|\mathbf{X}\mathbf{w}\|_2)\sqrt{\mathcal{G}_{\mathcal{C}}(\mathbf{w})/\mu} < 0 \tag{1}$$

~~is less than 0~~ at any $\mathbf{w} \in \mathcal{C}$, then $w_i^* = 0$, where $\mathbf{w}^*$ is the optimal solution to the optimization problem.[3]

Our goal is to employ this screening rule for linear regression, namely, when $f(\mathbf{X}\mathbf{w}) : \mathbb{R}^n \to \mathbb{R} = \frac{1}{n}(\mathbf{y} - \mathbf{X}\mathbf{w})^\top (\mathbf{y} - \mathbf{X}\mathbf{w})$. Since we want to guarantee privacy, we will determine the sensitivity of this calculation so we can add an appropriate amount of noise and ensure screening is differentially private. Specifically, we want to determine how much the value of $f$ can change between datasets $\mathbf{X}$ and $\mathbf{X}'$, where $\mathbf{X}'$ contains either one additional row or one removed row compared to $\mathbf{X}$. ~~Our goal is to determine the sensitivity of this calculation so we can add an appropriate amount of noise and ensure screening is differentially private.~~ We will conduct our analysis for the case where $\|\mathbf{x}_i\|_\infty \leq 1$, $\mathcal{C}$ is the $\lambda$-scaled $L_1$-ball in $\mathbb{R}^d$, and $|y_i| < \lambda$. These conditions are also required by the `DP-FW` optimizer (Talwar et al., 2015).

**Theorem 1.** *Under the conditions listed above, the sensitivity of Equation 1 when $f(\mathbf{X}\mathbf{w}) = \frac{1}{n}(\mathbf{y} - \mathbf{X}\mathbf{w})^\top (\mathbf{y} - \mathbf{X}\mathbf{w})$ is*

$$\frac{2\lambda}{n} + \frac{2\lambda^2}{n} + \frac{1}{n}(1 + \lambda)\sqrt{\frac{4\lambda^2/n}{1/n}}.$$

---

[2]Note that Raj et al. (2016) is written with semantics typical in the field of optimization - there are $d$ (d)atapoints and the goal is to optimize a vector with (n)umber of components $n$. In this paper, we use the standard statistical and machine learning conventions, in which there are $n$ (n)umber of datapoints and optimization is done over $d$ (d)imensions.

[3]Note that in this expression and in the Wolfe gap function, the gradients are taken with respect to the vector $\mathbf{u} = \mathbf{X}\mathbf{w} \in \mathbb{R}^n$. Since $\mathbf{u} \in \mathbb{R}^n$, $\nabla f(\mathbf{u}) \in \mathbb{R}^n$. This can be a source of confusion for those who are unfamiliar with the Wolfe gap function.

---

**Algorithm 1** `ADP-Screen`

---

**Require:** Privacy Parameters: $\epsilon_1 > 0$, $\epsilon_2 > 0$, $0 < \delta_1 \leq 1$, $0 < \delta_2 \leq 1$; Constraint: $\lambda > 0$; Iterations: $T$; Design Matrix: $\mathbf{X} \in \mathbb{R}^{n \times d}$ where $\|\mathbf{x}_i\|_\infty \leq 1$ for all $i \in \{1, \ldots, n\}$; Target: $\mathbf{y}$; $L_2$-Sensitivity: $\Delta_2$; Set of Iterations to Screen: $I\dot{\mathbf{i}}$.

1: $l \leftarrow |I|$
2: $\delta_{\text{iter}} \leftarrow \frac{\delta_2}{l+1}$
3: $\epsilon_{\text{iter}} \leftarrow \frac{\epsilon_2}{2\sqrt{2l \log(1/\delta_{\text{iter}})}}$
4: $\sigma^2 \leftarrow \frac{2\Delta_2^2 \log(1.25/\delta_{\text{iter}})}{\epsilon_{\text{iter}}^2}$
5: $\widehat{\mathbf{w}}^{(0)} \leftarrow$ Random Vector in $\{\mathbf{w} \in \mathbb{R}^d : \|\mathbf{w}\|_1 \leq \lambda\}$ ~~the $\lambda$-scaled $L_1$ Ball~~
6: **for** $t = 1$ to $T$ **do**
7:     $\widehat{\mathbf{w}}^{(t)} \leftarrow$ `DP-FW Step` $\left(\epsilon_1, \delta_1, \lambda, T, \mathbf{X}, \mathbf{y}, \widehat{\mathbf{w}}^{(t-1)}\right)$
8:     **if** $t \in I$ **then**
9:         **for** $i = 1$ to $d$ **do**
10:             $s_i \leftarrow |\mathbf{x}_{(i)}^\top \nabla f(\mathbf{X}\widehat{\mathbf{w}}^{(\mathbf{t})})| + (\mathbf{X}\widehat{\mathbf{w}}^{(\mathbf{t})})^\top \nabla f(\mathbf{X}\widehat{\mathbf{w}}^{(\mathbf{t})}) + L(\|\mathbf{x}_{(i)}\|_2 + \|\mathbf{X}\widehat{\mathbf{w}}^{(\mathbf{t})}\|_2)\sqrt{\mathcal{G}_\mathcal{C}(\widehat{\mathbf{w}}^{(\mathbf{t})})/\mu}$ ~~Equation 1 $(\mathbf{X}, \mathbf{y}, \widehat{\mathbf{w}}^{(t)}, \lambda)$~~
11:         $\mathbf{s} \leftarrow \mathbf{s} + \mathcal{N}(\mathbf{0}, \sigma^2\mathbf{I}_d)$
12:         $\widehat{\mathbf{w}}_j^{(t)} \leftarrow 0$ if $s_j < 0$ **for all** $j$
13: Output $\widehat{\mathbf{w}}^{(T)}$

---

*The sensitivity of releasing* $\mathbf{s} = \begin{bmatrix} s_1 & \ldots & s_d \end{bmatrix}^\top$ *is*

$$\frac{2\lambda\sqrt{d}}{n} + \frac{2\lambda^2\sqrt{d}}{n} + \frac{1}{n}\left(\sqrt{d} + \lambda\sqrt{d}\right)\sqrt{\frac{4\lambda^2/n}{1/n}}.$$

The proof is provided in the appendix.

## 4  `ADP-Screen`

Using the bounds for the $L_2$-sensitivity of the screening rule in Equation 1, we discuss our first attempt at implementing it into a differentially private linear regression training procedure. Prior to proceeding, we highlight that the method produced in this section is impractical for actual use; rather, the method demonstrates the critical problem of "overscreening" in private screening rules. We develop a method to address overscreening in Section 5.

Since our screening rule requires $L_1$-constrained optimization, we employ the private Frank-Wolfe algorithm developed in Talwar et al. (2015) to train regression models. To the best of our knowledge, this is the only differentially-private algorithm for $L_1$-constrained optimization. Additionally, the mean-squared error loss has Lipschitz constant 1 with respect to the $L_1$ norm, which satisfies the algorithm's requirement for $L_1$-Lipschitz loss functions.

Our algorithm is shown in Algorithm 1, abstracting away the steps required for the private Frank-Wolfe algorithm. Since our mechanism uses the advanced composition theorem with approximate differential privacy, we call this method `ADP-Screen`. We include the following guarantee for privacy:

**Lemma 2.** *Algorithm 1 is* $(\epsilon_1 + \epsilon_2, \delta_1 + \delta_2)$*-differentially private.*

*Proof.* First, note that each iteration of line 11 uses the Gaussian mechanism and is $(\epsilon_{\text{iter}}, \delta_{\text{iter}})$-differentially private (Dwork & Lei, 2009). By using the advanced composition theorem for approximate differential privacy, we can identify that $l$ compositions of $\epsilon_{\text{iter}}$ and $\delta_{\text{iter}}$ is $(\epsilon_2, \delta_2)$-differentially private (Dwork et al., 2006; 2010). The differentially private Frank-Wolfe algorithm guarantees the output of $T$ compositions of `DP-FW Step`s is $(\epsilon_1, \delta_1)$-differentially private. Following this, the basic composition theorem of approximate differential privacy guarantees that Algorithm 1 is $(\epsilon_1 + \epsilon_2, \delta_1 + \delta_2)$-differentially private. □

ADP-Screen did not produce good results, and as such, we include the experiments section demonstrating its performance in the appendix. Here, we include the main takeaways from our experiments, which will be used to develop a better-performing screening rule in the following sections.

1. ADP-Screen was unable to discriminate between true zero and true nonzero coefficients.

2. ADP-Screen screened too many coefficients per iteration, and since the Frank-Wolfe algorithm only updates one coefficient at a time, after a few iterations, the weight vector is only able to have up to a few nonzero coefficients which have not been screened to zero.

## 5 RNM-Screen

In this this section, we seek to improve the performance of ADP-Screen by modifying it in two ways:

1. We redistribute the total privacy budget to reduce the noisiness of private optimization.

2. We employ the report-noisy-max mechanism to reduce the screening noise to sub-$\mathcal{O}(\sqrt{d})$ and reduce overscreening.

### 5.1 Redistributing the Privacy Budget

Experiments with ADP-Screen indicated that noisy private optimization caused it to lose its ability to screen coefficients effectively. For this reason, we sought to reduce the amount of noise added during private optimization.

ADP-Screen relies on DP-FW for $L_1$-constrained optimization. To the best of our knowledge, this is the only algorithm which performs private $L_1$-constrained optimization, and as such, the only way to reduce the amount of noise added during private optimization is to redistribute the final privacy budget to reduce the noise for private optimization.

### 5.2 Report-Noisy-Min Mechanism

Experiments on ADP-Screen demonstrated that it overscreened coefficients, producing solution vectors which were too sparse to output useful predictions. To prevent overscreening, we chose to screen only one coefficient per iteration.

To achieve this, we use the report-noisy-max mechanism. The report-noisy-max mechanism is a technique for choosing an item with the highest score from a set given a function which computes the score. By finding the maximum of the negative values of the scores, the report-noisy-max mechanism can also be used to privately find the element with minimum score. We use the report-noisy-min technique to select the coefficient to screen every iteration.

To do this, after every iteration, we run Equation 1 on all coefficients. We use the report-noisy-min technique to select the coefficient which produced the smallest value in Equation 1 in a differentially private manner. We then screen this coefficient.

Our algorithm can be seen in Algorithm 2. We call the algorithm RNM-Screen due to its use of report-noisy-min in lines 9 and 10.

Using the report-noisy-min mechanism has another benefit: we are able to reduce the noise scale to sub-$\mathcal{O}(\sqrt{d})$ since the report-noisy-min mechanism only requires the sensitivity for a single coefficient of the output weight (Dwork et al., 2014).

We conclude this section with a proof of privacy:

**Lemma 3.** *Algorithm 2 is $(\epsilon_1 + \epsilon_2, \delta_1 + \delta_2)$-differentially private.*

---

**Algorithm 2** RNM-Screen

---

**Require:** Privacy Parameters: $\epsilon_1 > 0$, $\epsilon_2 > 0$, $0 < \delta_1 \leq 1$, $0 < \delta_2 \leq 1$; Constraint: $\lambda > 0$; Iterations: $T$; Design Matrix: $\mathbf{X} \in \mathbb{R}^{n \times d}$ where $\|\mathbf{x}_i\|_\infty \leq 1$ for all $i \in \{1, \ldots, n\}$; Target: $\mathbf{y}$

1: $\epsilon_{\text{iter}} \leftarrow \dfrac{\epsilon_2}{\sqrt{8T \log \frac{1}{\delta_2}}}$

2: $\Delta \leftarrow \frac{2\lambda}{n} + \frac{2\lambda^2}{n} + \frac{2\lambda(1+\lambda)}{n}$

3: $scale \leftarrow \frac{\Delta}{\epsilon_{\text{iter}}}$

4: $\widehat{\mathbf{w}}^{(0)} \leftarrow$ Random Vector in $\{\mathbf{w} \in \mathbb{R}^d : \|\mathbf{w}\|_1 \leq \lambda\}$~~the $\lambda$-scaled $L_1$ Ball~~

5: **for** $t = 1$ to $T$ **do**

6: $\quad \widehat{\mathbf{w}}^{(t)} \leftarrow$ DP-FW Step $\left(\epsilon_1, \delta_1, \lambda, T, \mathbf{X}, \mathbf{y}, \widehat{\mathbf{w}}^{(t-1)}\right)$

7: $\quad$ **for** $i = 1$ to $n$ **do**

8: $\quad\quad s_i \leftarrow |\mathbf{x}_{(i)}^\top \nabla f(\mathbf{X}\widehat{\mathbf{w}}^{(\mathbf{t})})| + (\mathbf{X}\widehat{\mathbf{w}}^{(\mathbf{t})})^\top \nabla f(\mathbf{X}\widehat{\mathbf{w}}^{(\mathbf{t})}) + L(\|\mathbf{x}_{(i)}\|_2 + \|\mathbf{X}\widehat{\mathbf{w}}^{(\mathbf{t})}\|_2)\sqrt{\mathcal{G}_{\mathcal{C}}(\widehat{\mathbf{w}}^{(\mathbf{t})})/\mu}$ ~~Equation 1 $(\mathbf{X}, \mathbf{y}, \widehat{\mathbf{w}}^{(t)}, \lambda)$~~

9: $\quad$ $\mathbf{s} \leftarrow \mathbf{s} + $ Laplace $(scale)$

10: $\quad$ $j \leftarrow$ Index of the smallest element of $\mathbf{s}$

11: $\quad$ **if** $s_j < 0$ **then**

12: $\quad\quad \widehat{\mathbf{w}}_j^{(t)} \leftarrow 0$

13: Output $\widehat{\mathbf{w}}^{(T)}$

---

*Proof.* First, note that lines 9 and 10 consist are the report-noisy-min mechanism with the noise scaled by the sensitivity of a single component, so for a single iteration they are $(\epsilon_{\text{iter}}, 0)$-differentially private (Dwork et al., 2014). Next, using the advanced composition mechanism for pure differential privacy, running the screening rule $T$ times produces an $(\epsilon_2, \delta_2)$-differentially private mechanism. Employing basic composition theorem of approximate differential privacy guarantees that Algorithm 2 is $(\epsilon_1 + \epsilon_2, \delta_1 + \delta_2)$-differentially private (Dwork et al., 2010). $\qquad\square$

### 5.3 Analyzing RNM-Screen's Screening Potential

To observe the behavior of screening only one coefficient per iteration, it is interesting to consider the case where coefficients which are updated and screened are uniformly chosen every iteration. Although this will almost certainly not hold during training, analyzing this case will provide us with insight into such a screening procedure. In this case, we have the following:

**Theorem 4.** *Let $\{I_1, I_2, \ldots, I_d\}$ be a set of indicator functions where $I_i$ takes value $1$ when the $i^{th}$ coefficient is nonzero after $T$ iterations of training. Then*

$$\lim_{T \to \infty} \mathbb{E}\left[I_1 + \cdots + I_d\right] = \frac{(d-1)^2 d}{(d-1)d + 1}$$

$$\lim_{d \to \infty} \mathbb{E}\left[I_1 + \cdots + I_d\right] = T - 1.$$

The proof is provided in the appendix.

From these limits, we can see that when RNM-Screen is run for infinite iterations, it approaches a solution with $\mathcal{O}(d)$ nonzero coefficients. From this result, we can infer that RNM-Screen slows the growth of the number of nonzero coefficients but is weak enough to prevent overscreening. The second limit is expected, as when the number of dimensions grows, it becomes increasingly unlikely that RNM-Screen will choose to screen coefficients which have been updated to be nonzero. This result is common among differentially private algorithms; for example, if the private Frank-Wolfe algorithm is run alone and the number of dimensions approaches infinity, its coefficient choices will approach uniformity.

Table 1: Average true positive rates, false positive rates, $F_1$ scores, and sparsities of 20 trials of `RNM-Screen` when run on synthetic data. True positives correspond to private nonzero coefficients which are truly nonzero. False positives correspond to private nonzero coefficients which are zero in the true solution.

|  | TPR | FPR | $F_1$ SCORE | SPARSITY |
|---|---|---|---|---|
| UNCORRELATED | 0.829 | 0.475 | 0.291 | 0.504 |
| CORRELATED | 0.957 | 0.281 | 0.444 | 0.371 |

## 6 Experiments with `RNM-Screen`

To test `RNM-Screen`, we analyzed its performance on two synthetic datasets and a number of real-world datasets. Results are shown below.

### 6.1 Synthetic Data

We began by using the synthetic dataset which Raj et al. (2016) used to test their nonprivate screening algorithm. Specifically, we generated 3000 datapoints in $\mathbb{R}^{600}$ from the standard normal distribution, and scaled the final dataset so $\|\mathbf{x}_i\|_\infty \leq 1$. We set the true weight vector $\mathbf{w}^*$ to be sparse with 35 entries of $+1$ and 35 entries of $-1$, and set $\mathbf{y} = \mathbf{X}\mathbf{w}^*$. Raj et al. (2016) demonstrated that the nonprivate screening rule listed in Equation 1 performs well on this dataset for linear regression. We verified this result, finding that using the nonprivate Frank-Wolfe optimizer with the nonprivate screening rule at every iteration produced a final weight vector in which nonzero components were only at the locations of nonzero components in the true weight vector and 55% of the true nonzero components were nonzero after training.

We also wanted to determine how correlated features affect the performance of the screening rule. To this end, we generated 3000 datapoints in $\mathbb{R}^{600}$ from $\mathcal{N}(\mathbf{0}, \Sigma)$ where $\Sigma_{ij} = 0.5^{|i-j|}$. We then scaled the data as above, the true weight vector remained the same, and $\mathbf{y}$ was found in the same way.

We tested the performance of two settings of `RNM-Screen` on each of these datasets. We ran `RNM-Screen` with $\epsilon_1 = 4.9$, $\epsilon_2 = 0.1$, $\delta_1 = \frac{1}{4000}$, $\delta_2 = \frac{1}{12000}$, $\lambda = 50$, and $T = 1000$.[4] Figure 1 and Table 1 demonstrate the results of this experiment. It is clear that `RNM-Screen` performs significantly better than `ADP-Screen` in that it is able to distinguish between the true nonzero and true zero coefficients in both cases and sets many more true zero coefficients to 0.

We define the $F_1$ score as

$$\frac{\#\text{Correct Nonzero Coeffs.}}{\#\text{Corr. Nonzero Coeffs.} + \frac{\#\text{Incorr. Nonzero Coeffs.} + \#\text{Incorr. Zero Coeffs.}}{2}}$$

and employ this as a quantitative metric to compare how well an algorithm is able to distinguish screening true nonzero and true zero coefficients.[5] We find that the $F_1$ scores of `RNM-Screen` were significantly better than those of `ADP-Screen` in both for both the uncorrelated and correlated cases, as measured by a nonparameteric sign test. This is not surprising, as from Figure 1 we can see that the number of true nonzero coefficients found by `ADP-Screen` is approximately 0, which produces an $F_1$ score of approximately 0.

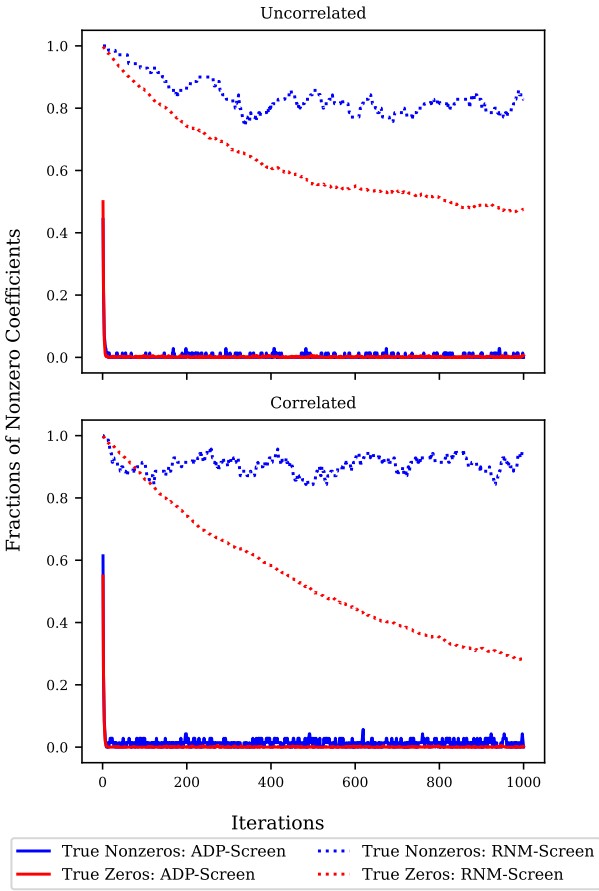

Figure 1: Comparing `RNM-Screen` to `ADP-Screen`. From these graphs, it is clear that `RNM-Screen` is able to distinguish between screening true nonzero coefficients and true zero coefficients, while `ADP-Screen` is generally unable to do this. However, in both cases, `RNM-Screen` does not achieve the same final sparsity level as `ADP-Screen`. Additionally, it is interesting to note that correlated features improves the performance of private screening rules.

## 6.2 Real-World Data

We also explored `RNM-Screen`'s performance on publicly available real-world datasets. We employed the Abalone, Housing, Body Fat, Pyrim, Triazines, Residential Buildings, Normalized Communities and Crime, and BlogFeedback datasets found in the LIBSVM and UCI Dataset Repositories (Harrison Jr & Rubinfeld, 1978; Behnke & Wilmore, 1974; Rafiei, 2018; Redmond, 2009; Buza, 2014). For the Residential Buildings dataset, our target variable was profit made on the sale of a property.[6] We applied the Yeo-Johnson transform to each dataset and scaled them so $\|\mathbf{x}_i\|_\infty \leq 1$ (Yeo & Johnson, 2000). We also applied a Yeo-Johnson transform to the the target $\mathbf{y}$. The dimensionality of each dataset can be found in Table 2.

---

[4]In our experiments, we chose to employ a high privacy budget for the optimization procedure to identify how private screening performs given a good optimizer but low privacy budget for screening. The experiments in the following sections will demonstrate that in many cases, `RNM-Screen` introduces minimal negative side-effects during optimization while inducing sparsity.

[5]This metric is inspired by the traditional $F_1$ score for binary classification. We choose to use it as a quantitative metric for measuring sparsity since, unlike traditional $L_0$ metrics for measuring sparsity, it rewards correct nonzero coefficients (coefficients which are nonzero in a nonprivate solution). It also rewards zero coefficients which are zero in a nonprivate solution and penalizes nonzero coefficients which are zero in a nonprivate solution.

[6]These datasets were chosen to have increasing dimensionality, but are still low-dimensional, with $n < d$. We did not employ high-dimensional datasets since running `DP-FW` on high-dimensional datasets is computationally challenging (Raff et al., 2024). Note, however, that sparsity on low-dimensional datasets with sufficiently high $d$ is still desirable since this can produce a more interpretable solution.

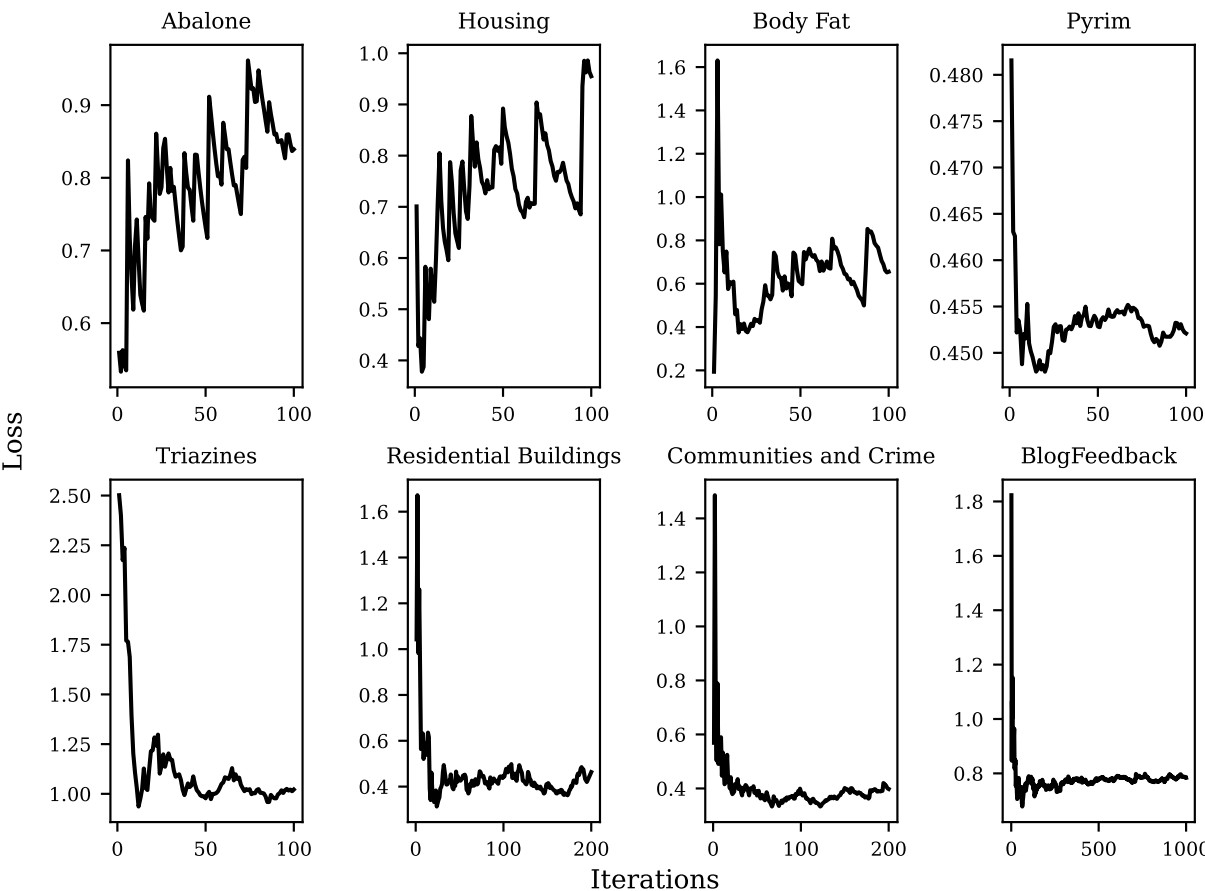

Figure 2: Mean squared error of `RNM-Screen` on real-world datasets. Decreasing loss on datasets with larger $d$ indicate that learning does occur in these instances. Since we are most interested in employing sparsity to build a generalizable and interpretable model on datasets with large $d$, this indicates that `RNM-Screen` may be useful for this task. Note that `DP-FW` does not produce sparse solutions, which is the purpose of this work. For this reason, it is excluded from this plot.

We chose to apply the Yeo-Johnson transform for a few reasons. First, the synthetic data tested above followed a Gaussian distribution, and to try and replicate its performance on real-world datasets, we transformed our data to have an approximately Gaussian distribution. Second, some of the datapoints and features in our real-world dataset contain outliers, and performing a linear scaling transformation to bound the $L_\infty$ norm of these points without the Yeo-Johnson transform would cause many of the values in the design matrices to be nearly zero. Finally, some of the datasets had outliers in their target variables which dominated mean-squared error loss. The Yeo-Johnson transform can combat this outcome. Note that we demonstrate results without the Yeo-Johnson transform in the appendix.

### 6.2.1  $F_1$ **Scores**

To measure how well `RNM-Screen` performed on these datasets, we compared its solutions to the weight vectors produced by scikit-learn's nonprivate LASSO optimizer (Pedregosa et al., 2011). We aimed to find a regularization strength for the nonprivate LASSO algorithm which produced approximately $\frac{d}{3}$ nonzero coefficients on a given dataset. We then used the $L_1$ norm of this output weight as $\lambda$ for `RNM-Screen`. For `RNM-Screen`, we set $\epsilon_1 = 4.9$, $\delta_1 = \frac{3}{4} \times \frac{1}{n}$, $\epsilon_2 = 0.1$, and $\delta_2 = \frac{1}{4} \times \frac{1}{n}$. For datasets with fewer datapoints, we ran the algorithm for fewer iterations as more iterations would increase the noise added in both the optimization and screening processes and unnecessarily corrupt the results.

Table 2: Average true positive rates, false positive rates, $F_1$ scores, sparsities, and $R^2$ scores of 20 trials of `RNM-Screen` when run on real-world dataset. We could not compare $R^2$ scores of `RNM-Screen` to `DP-FW` because the solutions of `DP-FW` are dense. True positives correspond to private nonzero coefficients which are nonzero in the nonprivate solution. False positives correspond to private nonzero coefficients which are zero in the nonprivate solution.

| | $n$ | $d$ | ITER. | TPR | FPR | $F_1$ | SPARSITY | $R^2$ |
|---|---|---|---|---|---|---|---|---|
| ABALONE | 4177 | 8 | 100 | 0.333 | 0.200 | 0.534 | **0.319** | 0.800 |
| HOUSING | 506 | 13 | 100 | 0.250 | 0.444 | 0.497 | **0.315** | 0.483 |
| BODY FAT | 252 | 14 | 100 | 0.600 | 0.444 | 0.333 | **0.357** | 0.653 |
| PYRIM | 74 | 27 | 100 | 0.556 | 0.556 | 0.445 | **0.506** | 0.688 |
| TRIAZINES | 186 | 60 | 100 | 0.474 | 0.488 | 0.377 | **0.508** | 0.851 |
| RESIDENTIAL BUILDINGS | 372 | 103 | 200 | 0.484 | 0.486 | 0.387 | **0.496** | 0.983 |
| COMMUNITIES AND CRIME | 1994 | 122 | 200 | 0.488 | 0.432 | 0.430 | **0.480** | 0.837 |
| BLOGFEEDBACK | 52397 | 280 | 1000 | 0.580 | 0.453 | 0.370 | **0.450** | 0.357 |

The results of this experiment are shown in Table 2. Of note, when computing the $F_1$ score, we defined the true zero and true nonzero coefficients with respect to the output of the nonprivate LASSO optimizer, not with respect to the optimal weight, as the optimal weight vector is unknown. Bolded values in Table 2 are significantly better than those produced by `DP-FW`, as measured by a nonparametric sign test. Note that we verified that running the Frank-Wolfe algorithm for more iterations on the smaller datasets produced little change in $F_1$ score while increasing the density of the solutions. This is why we chose to run `RNM-Screen` for fewer iterations on these datasets. For the larger BlogFeedback dataset, additional iterations did increase the density of the result but were accompanied with an increased $F_1$ score. This is why we ran this for more iterations.

Analyzing Table 2, it is clear that the results for $F_1$ scores are not ideal. For all datasets, `RNM-Screen` was unable to produce an $F_1$ score significantly better than the standard private Frank-Wolfe algorithm. Overall, `RNM-Screen` is significantly better at distinguishing between true nonzero and zero coefficients than `ADP-Screen`, but the results shown in Table 2 indicate that for real-world datasets, `RNM-Screen` does not effectively find the same nonzero coefficients as a LASSO solver.

### 6.2.2 Mean Squared Error

To determine whether `RNM-Screen` allowed any learning to occur, we tracked the mean squared error at every iteration on the above datasets. Results are shown in Figure 2.

Despite the disappointing results on $F_1$ scores, the mean squared error plots in Figure 2 indicate that learning still occurs on larger datasets despite the inaccurate choice of nonzero coefficients. We believe this is a valuable result, as it shows that the solutions produced by `RNM-Screen` may be useful in prediction. Indeed, research on the duality of differential privacy and algorithmic stability shows that differentially private algorithms do not seek to produce similar results to nonprivate algorithms but rather find algorithmically stable solutions which fit the data sufficiently well (Dwork & Lei, 2009).

We tested whether `RNM-Screen`'s nonzero coefficients served as an effective basis for the coefficients which it incorrectly set to zero to determine if the models produced by `RNM-Screen` have similar expressivity to nonprivate models. We did this by fitting an unregularized multivariate linear regression from the features of the dataset corresponding to `RNM-Screen`'s nonzero coefficients to the features of the dataset whose `RNM-Screen` set to zero but were nonzero in the nonprivate solution. We reported the $R^2$ scores of these regressions in Table 2. Higher values of $R^2$ scores indicate that `RNM-Screen`'s nonzero coefficients can better approximate the target features.

Table 2 indicates that for most datasets, the features which `RNM-Screen` chooses are able to approximate the unchosen features which the nonprivate method chooses reasonably well. This confirms our intuition that

Table 3: Comparing the $F_1$ scores and mean squared errors of 20 trials of the Oracle-$K$ privately optimized Frank-Wolfe with 20 trials of `RNM-Screen`. Results from the nonprivate Frank-Wolfe algorithm are included as `NP-FW` and demonstrate that without private optimization or pruning, the algorithm achieves higher $F_1$ scores and MSEs, as expected.

| | NONPRIVATE | | | | | | PRIVATE | |
| | NP-FW | | PRESELECT-$K$ FW | | ORACLE-$K$ FW | | RNM-SCREEN | |
| DATASET | $F_1$ | MSE | $F_1$ | MSE | $F_1$ | MSE | $F_1$ | MSE |
|---|---|---|---|---|---|---|---|---|
| ABALONE | 1.000 | 0.512 | 0.667 | 0.689 | **0.867** | **0.560** | 0.534 | 0.894 |
| HOUSING | 1.000 | 0.322 | 0.000 | 0.786 | **0.788** | **0.516** | 0.497 | 0.835 |
| BODY FAT | 0.833 | 0.025 | 0.400 | 0.246 | 0.320 | **0.368** | 0.333 | 0.689 |
| PYRIM | 0.889 | 0.439 | 0.571 | 0.759 | 0.339 | 0.451 | **0.445** | 0.451 |
| TRIAZINES | 0.829 | 0.610 | 0.467 | 0.655 | 0.337 | 0.979 | **0.377** | 0.979 |
| RESIDENTIAL BUILDINGS | 0.667 | 0.009 | 0.500 | 0.358 | 0.342 | 0.415 | **0.387** | 0.431 |
| COMMUNITIES AND CRIME | 0.800 | 0.277 | 0.540 | 0.373 | 0.400 | 0.382 | 0.430 | 0.395 |
| BLOGFEEDBACK | 0.868 | 0.489 | 0.546 | 0.489 | 0.320 | **0.736** | **0.370** | 0.822 |

although `RNM-Screen` does not choose the same nonzero coefficients as a nonprivate rule, it is still able to learn effectively from the coefficients it chooses.

In order to test whether the mean squared errors are good for their level of sparsity, we compared `RNM-Screen` to an Oracle-$K$ clip of the private Frank-Wolfe Algorithm. Specifically, for the Oracle-$K$ clip, we found a weight with the private Frank-Wolfe algorithm and then kept only the $K$ absolute largest coefficients nonzero, where $K$ is the number of nonzero coefficients which the nonprivate LASSO solver found. Note that this is not differentially private, as the value of $K$ is not private, but we employed it as a strong baseline. If `RNM-Screen` produces $F_1$ scores or mean squared errors similar to the Oracle-$K$ clip, we believe it performs well for a private sparse algorithm.

Results are shown in Table 3, which includes also includes two more nonprivate columns. The NP-FW column displays results from the nonprivate Frank-Wolfe algorithm without any feature selection, and the Preselect-$K$ FW column displays results from the nonprivate Frank-Wolfe algorithm after choosing the $K$ features with highest $L_1$ norm.[7] For all datasets with $d > 20$, `RNM-Screen` does not produce a significantly worse $F_1$ score than the Oracle-$K$ technique. For all but the BlogFeedback dataset, the datasets with $d > 20$ also do not have significantly worse mean squared errors. These results imply that `RNM-Screen` is an effective way to ensure privacy while producing a sparse solution. While it may not identify the same nonzero coefficients which a nonprivate LASSO solver identifies, it is still able to learn effectively while producing comparatively good $F_1$ scores and sparsity.

## 7 Open Problems

The experiments above present modest evidence that a screening rule can produce effective sparse solutions for linear regression under differential privacy. In this section, we highlight four open problems for further study.

First, we ask whether private screening can be adapted to effectively distinguish between the nonzero and zero coefficients of nonprivate solutions on real-world datasets, including high-dimensional ones. Future works to solving this problem may be empirical, perhaps improving `RNM-Screen`, but a more rewarding approach

---

[7]Note that we also tested using the nonprivate Frank-Wolfe algorithm with the screening rule, but this produced identical results to the NP-FW column. This is because nonprivate Frank-Wolfe optimization was started from the zero vector, and the algorithm did not update any features which the screening rule removed. Intuitively, this makes sense - the Frank-Wolfe algorithm only updated the coefficients which were beneficial in the final solution, and the screening rule did not make these coefficients zero.

would identify a utility bound for a private screening rule so it can be compared to other differentially private algorithms. A utility bound is especially useful in comparing differentially private randomized algorithms as it is not subject to random variations which occur when empirically comparing algorithms on datasets.

Second, we ask whether private screening rules can be employed in a similar approach to nonprivate screening rules. Screened coefficients from nonprivate screening rules will not be updated in future iterations by the optimizer, which improves the computational efficiency of optimization, especially for very high dimensional datasets. Unfortunately, since our screening rule is randomized, it is possible that a coefficient is screened incorrectly, so we continue to optimize for all coefficients despite screening, which does not improve the efficiency of optimization. An effective private screening rule which improves the computational efficiency of optimization would encompass both the sparsity and efficiency benefits of screening rules.

Third, we note that `RNM-Screen` requires two sets of privacy parameters, $(\epsilon_1, \delta_1)$ and $(\epsilon_2, \delta_2)$. Although we used a heuristic to set these parameters, consistent guidelines for splitting the privacy budget between these two parameters is unclear. Note that this issue arises in many works with multiple calculations that need to be privatized. For example, the original `Two-Stage` sparse regression technique proposed by Kifer et al. (2012) also required two sets of privacy parameters. However, in addition to further testing, to produce a private screening rule which can be employed for real-world data analysis, better guidelines for setting these privacy parameters is required.

Finally, we highlight a limitation to `RNM-Screen`. On low-dimensional datasets, every coefficient might be important, but `RNM-Screen` always screens one coefficient per iteration, which can still lead to overscreening. In contrast, nonprivate screening algorithms never screen coefficients which are important to the regression task. For this reason, it is important that `RNM-Screen` only be used on tasks with a sufficient number of features such that practitioners believe that a sparse solution should exist.

## 8 Conclusion

In this paper, we created `ADP-Screen`, the first differentially private screening rule. After showing that it overscreens coefficients and is unable to solve a simple regression problem with a synthetic dataset, we developed `RNM-Screen`. We showed that it performs better than `ADP-Screen` on synthetic data since it is actually able to distinguish between screening true zero and true nonzero coefficients. After testing on the synthetic datasets, we tested `RNM-Screen`'s performance on real-world datasets. We found that it produces good sparsity with decreasing mean-squared error on larger datasets. Additionally, $R^2$ scores indicate that the coefficients it chooses are able to model the coefficients it screened away during computation. Finally, we found modest evidence that the Yeo-Johnson transform improves `RNM-Screen`'s performance on real-world datasets. We believe that the algorithmic and empirical developments this work makes to the field of sparse regression with differential privacy makes it a valuable result to the exploration of future techniques for screening rules and sparse differentially private regression.

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

## A  Proof: Privatizing Iterative Screening

**Theorem 1.** *Under the conditions listed above, the sensitivity of Equation 1 when* $f(\mathbf{Xw}) = \frac{1}{n}(\mathbf{y} - \mathbf{Xw})^\top (\mathbf{y} - \mathbf{Xw})$ *is*

$$\frac{2\lambda}{n} + \frac{2\lambda^2}{n} + \frac{1}{n}(1 + \lambda)\sqrt{\frac{4\lambda^2/n}{1/n}}.$$

*The sensitivity of releasing* $\mathbf{s} = \begin{bmatrix} s_1 & \dots & s_d \end{bmatrix}^\top$ *is*

$$\frac{2\lambda\sqrt{d}}{n} + \frac{2\lambda^2\sqrt{d}}{n} + \frac{1}{n}\left(\sqrt{d} + \lambda\sqrt{d}\right)\sqrt{\frac{4\lambda^2/n}{1/n}}.$$

*Proof.* Let $\mathbf{u} = \mathbf{Xw}$. For linear regression, $f(\mathbf{u}) = \frac{1}{2n}(\mathbf{u} - \mathbf{y})^\top(\mathbf{u} - \mathbf{y})$, implying $\nabla f(\mathbf{u}) = \frac{1}{n}(\mathbf{u} - \mathbf{y})$ and $\nabla^2 f(\mathbf{u}) = \frac{1}{n}\mathbf{I}_n$. Therefore, from the definitions of Lipschitz smoothness and strong convexity, we can see that $f(\mathbf{u})$ is $\frac{1}{n}$-smooth and $\frac{1}{n}$-strongly convex with respect to $\mathbf{u}$.

By using the triangle inequality and the fact that the maximum of a sum is at most the sum of each element's maximum, we can bound the sensitivity of Equation 1 for linear regression by summing the sensitivity of each of its terms. These calculations are shown below. Assume without loss of generality that the difference between $\mathbf{X}$ and $\mathbf{X}'$ is the data-target tuple $(\mathbf{x}_0, y_0)$.

To bound the first term of Equation 1, note that

$$\max_{\mathbf{X},\mathbf{X}',y,y'} \left| \mathbf{x}_{(i)}^\top \nabla f(\mathbf{Xw}) - \mathbf{x}'^\top_{(i)} \nabla f(\mathbf{X}'\mathbf{w}) \right|$$

$$= \max_{\mathbf{x}_0,y_0} \left| x_{0i}\left[\frac{1}{n}\left(\mathbf{x}_0^\top\mathbf{w} - y_0\right)\right] \right|$$

$$\leq \frac{2\lambda}{n},$$

where the first simplification comes from expanding the inner products and the second follows directly from restrictions on the feasible set and norms of the input data. This means that the sensitivity of the first term for one value of $i$ is $\frac{2\lambda}{n}$. By the sensitivity principles of vectors, this means that releasing the values of the first term for all $i$ has $L_2$-sensitivity of $\frac{2\lambda\sqrt{d}}{n}$ (Dwork et al., 2014).

For the second term,

$$\max_{\mathbf{X},\mathbf{X}',y,y'} |(\mathbf{Xw})^\top \nabla f(\mathbf{Xw}) - (\mathbf{X}'\mathbf{w})^\top \nabla f(\mathbf{X}'\mathbf{w})|$$

$$= \max_{\mathbf{x}_0,y_0} \left| \mathbf{x}_0^\top\mathbf{w}\left[\frac{1}{n}\left(\mathbf{x}_0^\top\mathbf{w} - y_0\right)\right] \right|$$

$$\leq \frac{2\lambda^2}{n},$$

using the same logic as the previous calculation. This means its sensitivity is $\frac{2\lambda^2}{n}$ and its $L_2$-sensitivity is $\frac{2\lambda^2\sqrt{d}}{n}$.

To find the sensitivity of the third term, we note that the maximum of a product is bounded by the product of maximums. Given this, we find that

$$\max_{\mathbf{X},\mathbf{X}'} \left| \left\|\mathbf{x}_{(i)}\right\|_2 - \left\|\mathbf{x}'_{(i)}\right\|_2 \right|$$

$$\leq \max_{\mathbf{x}_0} \sqrt{|x_{0i}^2|}$$

$$\leq 1,$$

where the first simplification is derived from expanding the first and noting that the difference of square roots must be less than or equal to the square root of the absolute value of the difference in their squared terms. The same logic can be applied for

$$\max_{\mathbf{X},\mathbf{X}'} \left| \|\mathbf{Xw}\|_2 - \|\mathbf{X}'\mathbf{w}\|_2 \right|$$
$$\leq \max_{\mathbf{x}_0,y_0} \sqrt{\left| \left(\mathbf{x}_0^\top \mathbf{w}\right)^2 \right|}$$
$$\leq \lambda.$$

For the Wolfe gap function,

$$\max_{\mathbf{X},\mathbf{X}',y,y',\mathbf{z}\in\mathcal{C}} \left| (\mathbf{Xw} - \mathbf{Xz})^\top \nabla f(\mathbf{Xw}) - (\mathbf{X}'\mathbf{w} - \mathbf{X}'\mathbf{z})^\top \nabla f(\mathbf{X}'\mathbf{w}) \right|$$
$$= \max_{\mathbf{x}_0,y_0,\mathbf{z}\in\mathcal{C}} \left| \mathbf{x}_0^\top (\mathbf{w} - \mathbf{z}) \left[ \frac{1}{n} \left( \mathbf{x}_0^\top \mathbf{w} - y_0 \right) \right] \right|$$
$$\leq \frac{4\lambda^2}{n}.$$

Plugging in each of these calculations, the sensitivity of this term for a single $i$ is $\frac{1}{n}(1+\lambda)\sqrt{\frac{4\lambda^2/n}{1/n}}$ and the $L_2$-sensitivity for releasing all $i$ is bounded by $\frac{1}{n}\left(\sqrt{d} + \lambda\sqrt{d}\right)\sqrt{\frac{4\lambda^2/n}{1/n}}$. By the triangle inequality, this means the sensitivity of the screening rule for one $i$ is bounded by

$$\frac{2\lambda}{n} + \frac{2\lambda^2}{n} + \frac{1}{n}(1+\lambda)\sqrt{\frac{4\lambda^2/n}{1/n}} \tag{2}$$

and the total $L_2$-sensitivity for releasing all $i$ is bounded by

$$\frac{2\lambda\sqrt{d}}{n} + \frac{2\lambda^2\sqrt{d}}{n} + \frac{1}{n}\left(\sqrt{d} + \lambda\sqrt{d}\right)\sqrt{\frac{4\lambda^2/n}{1/n}}. \tag{3}$$

$\square$

## B  Experiments on `ADP-Screen`

To test whether `ADP-Screen` performs well in practice, we tested how it performed on linear regression. To do so, we used the synthetic dataset which Raj et al. (2016) used to test their nonprivate screening algorithm. Specifically, we generated 3000 datapoints in $\mathbb{R}^{600}$ from the standard normal distribution, and scaled the final dataset so $\|\mathbf{x}_i\|_\infty \leq 1$. We set the true weight vector $\mathbf{w}^*$ to be sparse with 35 entries of $+1$ and 35 entries of $-1$, and set $\mathbf{y} = \mathbf{X}\mathbf{w}^*$. Raj et al. (2016) demonstrated that the nonprivate screening rule listed in Equation 1 performs well on this dataset for linear regression. We verified this result, finding that using the nonprivate Frank-Wolfe optimizer with the nonprivate screening rule at every iteration produced a final weight vector in which nonzero components were only at the locations of nonzero components in the true weight vector and 55% of the true nonzero components were nonzero after training. We then ran the private `ADP-Screen` on this dataset for 1000 iterations with $\epsilon_1 = \epsilon_2 = 2.5, \delta_1 = \delta_2 = \frac{1}{6000}$, and $\lambda = 5$.

Figure 3 shows the results of this experiment when we implemented screening after every iteration, every $50^{\text{th}}$ iteration, and after the last iteration. It is clear that `ADP-Screen` is not able to discriminate between screening true zero and true nonzero coefficients in any of these cases. Additionally, when private screening is implemented too often, it screens too many coefficients, and since the Frank-Wolfe algorithm only updates one coefficient at a time, after a few iterations, the weight vector is only able to have up to a few nonzero coefficients which have not been screened to zero. To identify whether the private Frank-Wolfe algorithm or the private screening methods were causing these poor results, we ran the following two experiments:

(A)  We tested how well nonprivate screening performed using the private Frank-Wolfe algorithm with $\epsilon = 2.5$ and $\delta = \frac{1}{6000}$. We found that no matter how often we implemented the screening rule, no coefficients were screened from the solution.

(B)  We tested how well private screening performed using the nonprivate Frank-Wolfe algorithm with $\epsilon = 2.5$ and $\delta = \frac{1}{6000}$. We found that when screening every $50^{\text{th}}$ iteration, the screening rule would produce a final weight vector with nonzero components in approximately 10% of the true nonzero components and none of the true zero components. The results of this experiment when screening every iteration or only at the last iteration mimicked those found in the respective rows of Figure 1.

These experiments provide key insights into the results shown in Figure 3. Experiment A suggests that all of the screening occurring in Figure 3 arises from noise added to the screening rule. This is because without the screening's noise, no screening occurs. It also implies that the noise added for private optimization made it more difficult to screen coefficients, since when we tested completely nonprivate screening (without noisy optimization), the nonprivate screening rule worked well. Heuristically, this outcome may arise because noisy weights make the Wolfe gap function in Equation 1 very large, meaning that it overpowers the second term, which is the only term that can be negative and is thus essential to effective screening. We verified that the Wolfe gap function evaluates to smaller values when using nonprivate optimization. This result can be seen in Figure 4.

Experiment B indicates that the noise added in the private screening rule makes it much stronger that its nonprivate counterpart. This is observed by noting that without a noisy screening rule, screening at every iteration with the nonprivate Frank-Wolfe optimizer would not screen all the true nonzero components to zero, whereas with the noisy screening rule, almost all components are screened to zero after only a few iterations. This makes sense: in an iteration of `ADP-Screen`, many coefficients can be screened to zero, but at most one zero coefficient can become nonzero through the Frank-Wolfe update. When the coefficients chosen for screening are noisy, there is a high probability that each coefficient is selected to be screened at some point over many iterations, but only a few incorrectly screened coefficients can become nonzero from a Frank-Wolfe update.

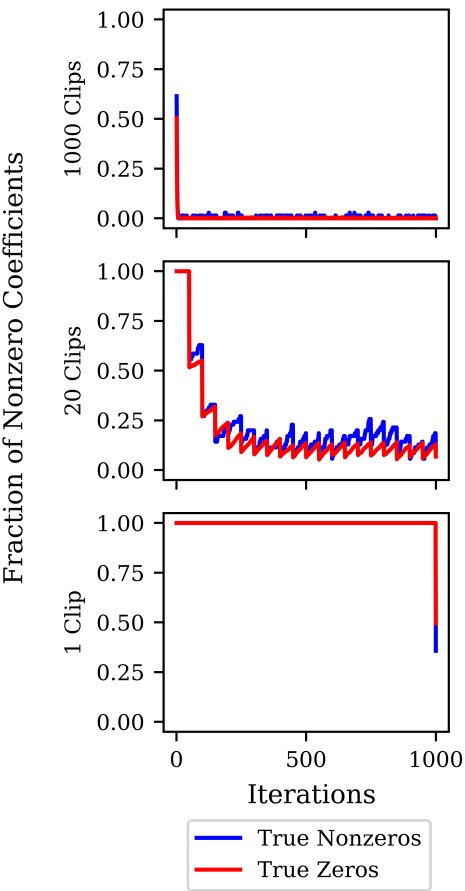

Figure 3: Testing `ADP-Screen` on a synthetic dataset. Values reported are the fraction of nonzero coefficients in the output of `ADP-Screen` which correspond to the indices of the true nonzero and true zero coefficients. The true nonzero and true zero coefficients are known from the dataset generation procedure.

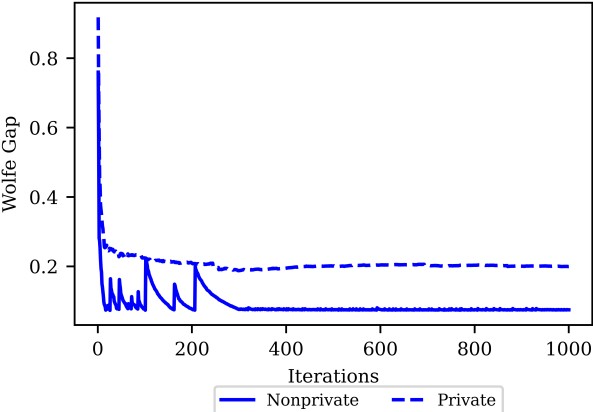

Figure 4: Comparing the Wolfe gap function for nonprivate and private optimization when nonprivate screening is applied at every iteration.

## C    Proof: Analyzing `RNM-Screen`'s Screening Potential

**Theorem 4.** *Let $\{I_1, I_2, \ldots, I_d\}$ be a set of indicator functions where $I_i$ takes value 1 when the $i^{th}$ coefficient is nonzero after $T$ iterations of training. Then*

$$\lim_{T \to \infty} \mathbb{E}\left[I_1 + \cdots + I_d\right] = \frac{(d-1)^2 d}{(d-1)d + 1}$$

$$\lim_{d \to \infty} \mathbb{E}\left[I_1 + \cdots + I_d\right] = T - 1.$$

*Proof.* We compute the expected value of the number of nonzero coefficients after training, $\mathbb{E}\left[I_1 + I_2 + \cdots + I_d\right]$. By the laws of expectation, this is equivalent to $\mathbb{E}\left[I_1 + I_2 + \cdots + I_d\right] = \mathbb{E}\left[I_1\right] + \mathbb{E}\left[I_2\right] + \cdots + \mathbb{E}\left[I_d\right]$. Since the coefficients are updated and screened uniformly, they are exchangeable, and thus this expectation simplifies to $d\mathbb{E}\left[I_1\right]$.

By the properties of Bernoulli random variables, the expectation of an indicator function is equal to the probability it is 1. We can see that $\mathbb{P}(I_1 = 1)$ equals

$\mathbb{P}$(Coef. 1 Never Screened)$\mathbb{P}$(Coef. 1 Updated)$+$

$\mathbb{P}$(Coef. 1 Last Screened At Ite. 1)$\mathbb{P}$(Coef. 1 Updated After Ite. 1)$+$

$\mathbb{P}$(Coef. 1 Last Screened At Ite. 2)$\mathbb{P}$(Coef. 1 Updated After Ite. 2)$+$

$\cdots+$

$\mathbb{P}$(Coef. 1 Last Screened At Ite. T)$\mathbb{P}$(Coef. 1 Updated After Ite. T).

This can be calculated as

$$\left(\frac{d-1}{d}\right)^T \left(1 - \left(\frac{d-1}{d}\right)^T\right) + \sum_{i=1}^{T} \frac{1}{d}\left(\frac{d-1}{d}\right)^{T-i}\left(1 - \left(\frac{1}{d}\right)^{T-i}\right)$$

where the summation breaks each event up into the $\frac{1}{d}$ probability that coefficient 1 was screened on iteration $i$, the $\left(\frac{d-1}{d}\right)^{T-i}$ probability that coefficient 1 was not screened after iteration $i$, and the $\left(1 - \left(\frac{1}{d}\right)^{T-i}\right)$ probability that coefficient 1 was updated after iteration $i$. Therefore $d\mathbb{E}\left[I_1\right]$ equals

$$d\left(\frac{d-1}{d}\right)^T \left(1 - \left(\frac{d-1}{d}\right)^T\right) + \sum_{i=1}^{T} \left(\frac{d-1}{d}\right)^{T-i}\left(1 - \left(\frac{1}{d}\right)^{T-i}\right).$$

This expression holds only when the assumptions in this section are satisfied. However, as the number of iterations approaches infinity, the scale of differentially private noise increases unbounded, and will thus dominate any finite gradient. This means that as $T \to \infty$, choosing coefficients to update and screen approaches uniformity. Additionally, as the number of dimensions approaches infinity, the probability of choosing a specific coefficient to update or screen approaches $\frac{1}{d}$. Thus, we compute the limits of $d\mathbb{E}\left[I_1\right]$ under these limits:

$$\lim_{T \to \infty} d\mathbb{E}\left[I_1\right] = \frac{(d-1)^2 d}{(d-1)d + 1},$$

$$\lim_{d \to \infty} d\mathbb{E}\left[I_1\right] = T - 1.$$

$\square$

## D  Ablating the Yeo-Johnson Transform

We wanted to determine whether the Yeo-Johnson transform had a significant impact on `RNM-Screen`'s performance. For this reason, we chose to run an ablation study in which we did not apply the Yeo-Johnson transform to the features or targets. Instead, we scaled datapoints so $\|\mathbf{x}_i\|_\infty \leq 1$ and $|y_i| \leq 1$. We ran this ablation study for each of the real-world datasets in the previous section for the same number of iterations as listed in Table 2.

Table 4 and Table 5 list the results of this experiment. Note that it is impossible to directly compare the mean-squared error values to previous experiments because the scaling of the target variables is different.

From Table 4, it is clear that as before, running `RNM-Screen` on this dataset does not produce significantly better scores than the standard private Frank-Wolfe algorithm, but it does significantly improve the sparsity. However, for all datasets except BlogFeedback, Table 4 has $R^2$ scores that are worse than Table 2. This indicates that the Yeo-Johnson transform allows our chosen features to better approximate unchosen important features when building a model.

Table 5 demonstrates a significant difference between the results of the transformed and untransformed data. With the transformed data, Table 3 indicates that `RNM-Screen` outperforms Oracle-$K$ FW on larger datasets with respect to the $F_1$ score. However, in Talble 5, this does not happen for any dataset. This may indicate that transforming the data makes it easier for `RNM-Screen` to distinguish between true zero and true nonzero coefficients, which thus improves its performance with respect to the $F_1$ score.

Table 4: Average $F_1$ scores, sparsities, and $R^2$ values of 20 trials of `RNM-Screen` when run on real-world datasets which have not been transformed with the Yeo-Johnson transform.

|  | $F_1$ | Sparsity | $R^2$ |
|---|---|---|---|
| Abalone | 0.450 | **0.113** | 0.109 |
| Housing | 0.525 | **0.200** | 0.062 |
| Body Fat | 0.447 | **0.204** | 0.526 |
| Pyrim | 0.451 | **0.441** | 0.582 |
| Triazines | 0.412 | **0.497** | 0.808 |
| Residential Buildings | 0.372 | **0.240** | 0.856 |
| Communities and Crime | 0.433 | **0.505** | 0.821 |
| BlogFeedback | 0.367 | **0.439** | 0.383 |

Table 5: Comparing the $F_1$ scores and mean squared errors of 20 trials of the Oracle-$K$ privately optimized Frank-Wolfe with 20 trials of `RNM-Screen` on data which has not been processed with the Yeo-Johnson transform.

|  | Nonprivate | | | | Private | |
|---|---|---|---|---|---|---|
|  | NP-FW | | Oracle-$K$ FW | | RNM-Screen | |
| Dataset | $F_1$ | MSE | $F_1$ | MSE | $F_1$ | MSE |
| Abalone | 1.000 | 0.012 | **0.667** | **0.014** | 0.450 | 0.010 |
| Housing | 0.889 | 0.018 | **0.788** | **0.051** | 0.525 | 0.159 |
| Body Fat | 1.000 | 0.020 | **0.550** | **0.124** | 0.447 | 0.555 |
| Pyrim | 0.941 | 0.019 | **0.583** | **0.182** | 0.451 | 0.301 |
| Triazines | 0.872 | 0.024 | 0.424 | 0.108 | 0.412 | 0.095 |
| Residential Buildings | 0.061 | 2.120 | 0.411 | 1.711 | 0.372 | 1.820 |
| Communities and Crime | 0.469 | 0.020 | **0.400** | 0.033 | 0.433 | 0.034 |
| BlogFeedback | 0.021 | 0.023 | **0.349** | 0.022 | 0.367 | **0.004** |

