# OpenReview forum: "Differentially Private Iterative Screening Rules for Linear Regression"
_TMLR — Rejected by TMLR_

### Review · Reviewer_Kq1S · 2024-03-25

**Summary Of Contributions:**

This paper studies differentially private screening of model parameters, i.e., those not significantly different from 0 can be made 0.

**Audience:**

Yes

**Claims And Evidence:**

No

**Requested Changes:**

A journal article doesn't have strong space limitations as in a conference paper.  TMLR attaches more importance to correctness than to significance, so rigorously developing all definitions and derivations, if needed in appendix, is essential.  So please rewrite the paper and make sure that each step is clear, rigorous and sound, and that you don't use concepts which you didn't define already.

**Strengths And Weaknesses:**

### General

Deriving a rule to select features which should have a non-zero coefficient in a linear regression model is an interesting question.
However, the current paper, as I already pointed out in my IJCAI-2024 review, the text very rapidly plunges into mathematical calculations without giving explanation or motivation for these calculations. Later on, quite a few sentences don't seem consistent / rigorous.  I provide below another small sample of issues which I encounter while reading the paper.  In conclusion, the paper remains unclear, the results remain unverifiable,

### Details

#### Section 3.

* There is no formal problem statement apart from the "light" problem statement in Section 1.  Section 2 starts without defining all used notations and concepts, e.g., it doesn't define the function f, nor the design matrix $X$.
* Mismatching dimensions.
    * $w$ is the weight vector and $X$ the design matrix.  The use of $Xw$ suggests that $w$ is a column vector and the rows of $X$ are the $x_i$ in the problem statement of Section 1.  We have $X\in\mathbb{R}^{n\times d}$.  $x_{(i)}$ is said to be the i-th column of $X$, so $x_{(i)}\in\mathbb{R}^n$ is a column vector.   While not explicitly specified, I guess gradients are taken with respect to the weight vector $w$, so $\nabla f(Xw) = \nabla_w f(Xw) \in \mathbb{R}^d$.  The first equation of this section multiplies $x_{(i)}^\top \nabla f(Xw)$ but the dimensions $1\times n$ and $d\times 1$ don't match.
    * Similarly, with $X\in\mathbb{R}^{n\times d}$ and $w\in\mathbb{R}^d$, we get $Xw\in\mathbb{R}^n$, and the product $(Xw)^\top \nabla f(Xw)$ is problematic if $\nabla f(Xw) \in\mathbb{R}^d$.
    * It seems one of my assumptions should be wrong.  There are two options: (a) $f$ is not real-valued but $f\in\mathbb{R}^n$.  In that case we would have $\nabla f(Xw)\in\mathbb{R}^{n\times d}$.  (b) gradients should not be taken with respect to $w$ (even though we are optimizing with respect to $w$), we then get $f:\mathbb{R}^n \to \mathbb{R}$ and $\nabla f(Xw) = (\nabla f) (Xw) \in \mathbb{R}^n$.
    * As the result of the summation seems to be real-valued, option (b) seems more likely.  Hence, I conjecture $f:\mathbb{R}^n\to \mathbb{R}$.
    * To verify the above guesses, the provided reference [Raj et al 2016] is not of great help.  First, the reference doesn't say which lemma/corollary/theorem of the cited paper is used here.  Second, the cited paper is a non-peer-reviewed arxiv paper so we have no guarantee the result is correct.  Third, the cited paper seems to interchange $n$ and $d$ compared to the current paper, which makes things especially tedious for verifying matching dimensions.  Please note that in a self-contained paper it should not be necessary to first read another paper in order to understand the first few lines.
* I guess that "is less than 0" means a strict comparison, i.e., "... < 0 ".
* Theorem 1 refers to "Equation 1" but in the PDF I downloaded I the equations don't seem to be numbered.
    * The paper does not define "sensitivity".  Typically, sensitivity is defined as $max_{x,y} |f(x)-f(y)|$, but given the above confusion about dimensions and given that [Raj 2016] uses primal/dual problems, many options exist here: Do you mean that the sensitivity is $\max_{x,y\in \mathbb{R}^n} |f(x)-f(y)|$, or rather $\max_{v,w\in\mathbb{R}^d} |f(Xv)-f(Xw)|$, or rather $\max_{X,Y\in\mathbb{R}^{n\times d}} |f(Xw)-f(Yw)|$ ?
* In appendix A, the first step of the proof specializes $f$ into a linear regression loss function.  Hence, the proof of Theorem 1 only holds for linear regression, and the Theorem is not proven in general.  If it is true in general, please provide a complete proof.  If it is not true in general, please limit the claim to linear regression loss functions.
    * In fact, not all treatments of linear regression set $f(u) = (u-y)^\top (u-y)/2n$, some also use $f(u) = (u-y)^\top (u-y)/n$ or $f(u) = (u-y)^\top (u-y)$ (it doesn't make a difference for the proof, but formally it is not correct to state that all linear regression approaches use this function), and there is even (regularized) linear regression using $f(u) = (u-y)^\top (u-y)/2n + \gamma \|w\|$.
    * In the unnumbered 3-line equation for the first term, is there any difference between the last expression on the first line and the expression on the second line?

#### Section 4:

* Algorithm 1 is unclear
    * algorithm 1 has a parameter "Iterations to screen" $\mathbf{i}$.  Please define what is an "iteration to screen".  Apparently there exists an (undefined) function $length$ that can be applied to $\mathbf{i}$,
    * I guess at line 5 you use an uniform probability distribution, and you didn't define the "$\lambda$-scaled $L_1$ ball" to be $\\{x\in\mathbb{R}^d| \\|x\\|_1\le \lambda\\}$ (well, assuming you mean the closed ball rather than the open ball which would be $\\{x\in\mathbb{R}^d| \\|x\\|_1 <\lambda\\}$.
    * at line 8, we have $t\in\mathbf{i}$.  While this suggests that $\mathbf{i}$ could be a set, this wouldn't be consistent with the existence of a function $length$.  If one would need for example the number of elements of $\mathbf{i}$, one could simply write $|\mathbf{i}|$.  A function $length$ is more typical for strings.
    * line 9 is unclear;  It applies the non-existing Equation 1.  I was guessing that Equation 1 may be the first equation in Section 3, but that equation does not contain an y nor $\lambda$, so it would be unclear what variable in that expression to substitute for $y$ and $\lambda$.  It would be much easier to just define a function in Equation 1, number the equation, and here refer to the function name rather than the equation number.

---

### Review · Reviewer_6AAd · 2024-04-14

**Summary Of Contributions:**

The goal of the paper is to provide differentially private screening rules for regression. The authors develop differentially private algorithms to decide which features should and should not be used in linear regression while keeping good performance and satisfying notions of sparsity.

**Audience:**

Yes

**Claims And Evidence:**

Yes

**Requested Changes:**

See weaknesses above

**Strengths And Weaknesses:**

Strengths:
- I believe the main contribution of the paper is interesting and valuable
- Section 6 provides an algorithm for the problem at hand that seems to perform well in practice, as per the experiments of Section 7. In particular, Figure 3 highlights that the fraction of zeroed-out true non-zero coefficients remain small, while the number of zeroed-out true zero coefficients decreases as the number of iterations increasing. While not perfect, the algorithm seem to keep all the ``relevant'' features while getting rid of a significant fraction of the non-relevant features and providing useful sparsity guarantees, unlike previous work.

Weaknesses:
- I think it takes a bit of time for the paper to get to the point. The main algorithm that performs well is in Section 6, and Section 5 is dedicated to an algorithm that is a useful building block but has poor performance in practice. I wonder if it would make sense to have Section 5 in Appendix but refer to it in Section 6 to justify the main algorithm.
- I do find the F1 scores a bit hard to interpret, and Figure 3 seems to have a true positive/true negative interpretation of the results instead that may be a bit cleaner. Could the authors report TP/TN rates on top of F1 scores?
- Can the authors provide baseline comparisons to the case where there is no initial private feature selection and the full information is used, to further highlight the performance of their DP feature selection process? (I believe as per Figure 3 that since they do not get rid of relevant features, it should perform almost as good as the version without DP feature selection, but it would be useful to compare still)

My overall evaluation here is that the paper does something interesting and I am in favor of it being accepted at TMLR, but I think a few more experimental baselines and scores as per the above "weaknesses" could be useful

---

### Review · Reviewer_GgGS · 2024-04-30

**Summary Of Contributions:**

This paper tackles the problem of designing privacy-preserving feature screening rules for $L_1$ regularized linear models. More specifically, the paper aims at enhancing the privacy-preservation of the screening rule presented in (Raj et al. 2016) and initially designed to sparsity the parameter updates of the Frank-Wolfe algorithm for high-dimensional learning problems. This screening methods essentially relies on verifying the positivity of a data-related criteria for every feature $i$. Let us denote this criteria by $\phi(x_{(i)})$, where $x_{(i)}$ is the $i$ th column of the data matrix. If $\phi(x_{(i)}) < 0$ then the parameter associated with the $i$ th feature should be zeroed-out.
The specific notion of privacy being used here is the well-know concept of differential privacy. The focus of this paper is related to the fact that the Frank-Wolfe’s algorithm already has a differentially private implementation, but (to the best of my knowledge) the associated screening rule has not been considered so far. As the screening rule from (Raj et al. 2016) also makes some calls to the data matrix, if we aim to use it on top of the DP Frank-Wolfe Algorithm, we need to also make it differentially private. In short, the paper presents two solutions.

- The first method is based on applying the Gaussian mechanism on top of $\phi(x_{(i)})$, i.e. injecting Gaussian noise directly on $\phi(x_{(i)})$, before checking its positivity. However, this method is too simplistic as the Gaussian noise essentially destroys the information contained in $\phi(x_{(i)})$ and the screening rule « overscreens », making the final method quite inaccurate.

- The second method is based on applying the report noisy max algorithm on the ensemble $\{ \phi(x_{(i)}) \}_{i \in [d]}$ (where $d$ is the number of features). Report noisy max is the gold standard differentially private algorithm for argmax selection.  In this version, since the algorithm only provides a single feature to zero-out, the method is much less aggressive and better preserves the final accuracy of the model.

**Audience:**

No

**Claims And Evidence:**

No

**Requested Changes:**

Overall, this article seems to me too preliminary to be published as it is. I think points 1) and 2) are important concerns that need to be addressed in depth. Point 3) is more minor.

**Strengths And Weaknesses:**

I feel that studying differentially private screening rules is an interesting topic. In this sense, the article focus seems quite relevant, even if it studies a rather narrow sub-question in this topic. Indeed, the article only focuses on a very specific algorithm and does not really justify why this algorithm is the best choice for implementing a private screening rule.

Furthermore, from my point of view, the paper is well written and the mathematical presentation (i.e. the formalism and presentations of the algorithms and claims) of the paper is sufficiently clear and formal. However, the paper, in its current form, has several weaknesses which, in my opinion, need to be addressed.


**1) Related work and comparison with earlier work (or simple benchmarks).** Current related work seems to me to be in very scarce. I understand that specific screening rules for $L_1$ regularized linear models with differential privacy may not yet exist, but it would seem rather odd to me that no algorithms with differential privacy have been proposed for vector sparification, thresholding functions or vector ranking. All these techniques would be appropriate solutions for feature vector screening and I think they should be mentioned and compared with. Similarly, in the experimental section, I would have liked to see more comparisons, if not with previous work (based on the remark above), at least with dummy benchmarks. What I mean by this is that there are dummy filtering rules that do not have access to the data, such as deleting features with the smallest $L_1$ norm. These rules may not be the best alternatives in general, but they are private by design (therefore require no additional noise injection) and seem reasonable enough to be compared with the proposed screening rules.

**2) The privacy guarantees in the paper lack sufficient justification.** In my opinion, even when an algorithm's privacy guarantees are easy to demonstrate, they should always be accompanied by a formal statement and associated proof. In the current submission, neither the first nor the second solution are accompanied by such statements/proof. Furthermore, the second algorithm, which is not based on the same privacy mechanism (therefore does not have the same sensitivity), does not include a proper algorithm sensitivity analysis. I think these analysis should be provided clearly. I also think that the privacy analysis presented here is rather loose (for example, it uses a loose version of the differential privacy composition theorem). Finally, the reference privacy level used in the article seems rather high to me ($\epsilon = $5$ is what is usually used for much more complex models, for example in deep learning). I think this level of privacy needs to be better explained and justified.

**3) Finally, I do not really see the point of presenting Algorithm 1.** From a methodological point of view, it is a direct application of the Gaussian mechanism. From an implementation point of view, it does not seem to work (as might be expected from the sensitivity's dependency on dimensions and the aforementioned loose privacy calculations). All in all, this algorithm does not offer much to the reader in my opinion. I would advise pushing the observation that a direct application of the Gaussian mechanism does not work to the appendix and focusing on the presentation of Algorithm 2 in the main document.

---

### Author Response · Authors · 2024-04-05
**Modified Manuscript has a Number of Clarifications**

Dear Reviewers, since one of the reviewers responded to our submission quickly, we have chosen to upload a modified manuscript based on their review. We believe that these modifications address many potential misunderstandings of our paper, and encourage you to read the updated manuscript. Changes are highlighted in blue.

---

### Decision · Action_Editor_B6Hf · 2024-06-10

**Recommendation:** Reject

**Comment:**

Despite of the rebuttal discussions and modifications made to the paper, both reviewers Kq1S and GgGS are leaning to reject. I share especially the concern of the reviewer Kq1S regarding the clarity and correctness and I am thus recommending reject.

I think the content and topic seems interesting and would fit very to TMLR. The paper just does not seem mature yet. In case of resubmission, I would recommend going carefulyl through the comments by reviewers GgGS and Kq1S (see also comments above). Especially, you should pay attention to the proof of Thm. 1 and its presentation. The algorithm "DP-FW Step", i.e. the DP Frank-Wolfe algorithm seems like a central step in both of the methods (Alg. 1 and Alg. 2). I think it should be explicitly given somewhere in the paper.

**Audience:**

Topic-wise and content-wise the paper would fit TMLR very well.

**Claims And Evidence:**

This was initially the concern of reviewer Kq1S and also later the concern of reviewer GgGS: the claims are not supported by clear and convincing evidence. While the idea seems nice, making DP versions of the screening rules that help in sparisfying model for linear regresion via the Gaussian mechanism, especially the sensitivity analysis does not yet seem to meet the bar for acceptance.

I could not myself  follow the proof of Thm. 1 either. Especially I do not see why the gradient bound $L$ does not appear there in case the sensitivities of those prodcut expressions are being bounded (see e.g. the first inequality in the proof of Thm. 1). All in all, the proof is hard to read, I could alson not understand how the strong convexity parameter derived in the beginning of the proof is used later on. Related to the proof of Thm. 1 and the notation, I would recommend carefully going through the comments of reviewer Kq1S.

Also, as pointed out by reviewer GgGS, the privacy analysis could be very easily improved by considering, e.g., RDP/zCDP analysis or privacy loss random variable based analysis for the Gaussian mechanism (see, e.g., Balle and Wang, 2018, "Improving the Gaussian Mechanism for Differential Privacy: Analytical Calibration and Optimal Denoising"). But more critical thing related to the privacy analysis, I think, is that experimental results are only shown for the total budget $\varepsilon=5$, which is rather big as pointed out by reviewer GgGS.

**Resubmission Of Major Revision:**

The authors may consider submitting a major revision at a later time.